# Liposome-Micelle-Hybrid (LMH) Carriers for Controlled Co-Delivery of 5-FU and Paclitaxel as Chemotherapeutics

**DOI:** 10.3390/pharmaceutics15071886

**Published:** 2023-07-04

**Authors:** Md. Musfizur Hassan, Bilquis Romana, Guangzhao Mao, Naresh Kumar, Fabio Sonvico, Pall Thordarson, Paul Joyce, Kristen E. Bremmell, Timothy J. Barnes, Clive A. Prestidge

**Affiliations:** 1School of Chemistry, The Australian Centre for Nanomedicine, The University of New South Wales, Sydney, NSW 2052, Australia; 2School of Chemical Engineering, The University of New South Wales, Sydney, NSW 2052, Australia; 3Clinical and Health Sciences, University of South Australia, Adelaide, SA 5000, Australia; 4Department of Food and Drug, University of Parma, 43124 Parma, Italy

**Keywords:** paclitaxel, 5-fluorouracil, liposomes, micelles, liposome-micelle hybrid, cancer cell uptake

## Abstract

Paclitaxel (PTX) and 5-fluorouracil (5-FU) are clinically relevant chemotherapeutics, but both suffer a range of biopharmaceutical challenges (e.g., either low solubility or permeability and limited controlled release from nanocarriers), which reduces their effectiveness in new medicines. Anticancer drugs have several major limitations, which include non-specificity, wide biological distribution, a short half-life, and systemic toxicity. Here, we investigate the potential of liposome-micelle-hybrid (LMH) carriers (i.e., drug-loaded micelles encapsulated within drug-loaded liposomes) to enhance the co-formulation and delivery of PTX and 5-FU, facilitating new delivery opportunities with enhanced chemotherapeutic performance. We focus on the combination of liposomes and micelles for co-delivery of PTX and 5_FU to investigate increased drug loading, improved solubility, and transport/permeability to enhance chemotherapeutic potential. Furthermore, combination chemotherapy (i.e., containing two or more drugs in a single formulation) may offer improved pharmacological performance. Compared with individual liposome and micelle formulations, the optimized PTX-5FU-LMH carriers demonstrated increased drug loading and solubility, temperature-sensitive release, enhanced permeability in a Caco-2 cell monolayer model, and cancer cell eradication. LMH has significant potential for cancer drug delivery and as a next-generation chemotherapeutic.

## 1. Introduction

Clinical cancer chemotherapy with paclitaxel (PTX) is currently limited due to its low oral bioavailability (<10%) resulting from its low aqueous solubility and dissolution kinetics [1,2], poor intestinal permeability [3], and first-pass hepatic metabolism [4,5]. PTX is also a P-glycoprotein (P-gp) substrate, resulting in efflux from the intestinal tract and limited efficacy against drug resistance [6,7,8]. Overexpression of efflux pumps, such as P-gp, is one of the major causes of multi-drug resistance (MDR). In breast cancer, P-gp-related drug resistance has been reported to occur in approximately 40% of breast cancer cells [9,10]. Other mechanisms contributing to MDR include reduced drug uptake, resistance to drug-related apoptosis, and the ability to repair DNA damage [11,12]. Most of the frontline chemotherapeutics, such as PTX, cisplatin, and doxorubicin, are P-gp substrates and induce P-gp overexpression with associated MDR [13]. Thus, effective oral administration of PTX is challenging, and intravenous (i.v.) administration is the clinically used dosage route [1].

The pyrimidine analogue fluorouracil (5-FU) has broad antitumor action, often providing synergistic activity with other anticancer drugs [14], e.g., a modified form of 5-FU can be used in association with PTX to achieve optimal therapeutic benefits against drug-resistant cancer [15]. 5-FU is sparingly soluble in water and slightly soluble in alcohol; hence, it has low bioavailability [16]. Furthermore, 5-FU has a short plasma half-life after i.v. bolus administration, requiring high doses that lead to severe gastrointestinal and cardiovascular toxicity along with the potential development of drug resistance by tumor cells [17,18]. The physicochemical and biopharmaceutical characteristics of PTX and 5-FU significantly limit their use as oral formulations, and desirable i.v. combination products are not available [1,19]. Hence, new drug delivery systems for 5-FU and PTX (and their combinations) are required to achieve better therapeutic efficacy with fewer side effects [20].

Nanomedicine approaches have emerged to improve solubility, reduce side effects, and enable targeted delivery of anticancer drugs [21,22,23]. Liposomes can incorporate drug candidates either within their lipophilic bilayer or hydrophilic core and provide advantages for improving drug stability, plasma half-life, and modulating toxicity, as reviewed previously [24]. Liposomes have been successfully translated to the clinic; e.g., liposomal doxorubicin (Doxil^®^), the first FDA-approved nanomedicine, and a liposomal PTX (Lipusu^®^) were recently approved by the FDA [25,26]. Thermal enhancement of drug cytotoxicity is also being established for improved chemotherapy; e.g., thermosensitive doxorubicin liposome (ThermoDox^®^), in combination with mild hyperthermia, was reported to be significantly more effective than the free drug in treating human squamous cell carcinoma xenografts [27,28]. Riganti et al. reported that liposomal doxorubicin effectively inhibits P-gp and reverses doxorubicin resistance in drug-resistant 126 HT29-dx cells [29]. Resveratrol and 5-FU coencapsulated in PEGylated liposomes improved chemotherapeutic efficacy against head and neck squamous cell carcinoma [30], and liposomal 5-FU was found to increase the accumulation of the drug in tumor tissue [22]. However, liposomes suffer from poor drug loading and a limited ability to control release.

Micelle-based encapsulation and delivery systems can increase the solubility of poorly water-soluble drugs, offer controlled release, and enhance circulation [31,32,33]. A micellar formulation of cyclosporine was approved for ocular application by the FDA in 2018 [34]. It is also noteworthy that phospholipid liposomes [35] and d-α-tocopheryl polyethylene glycol 1000 succinate (TPGS) micelles can block P-gp transport in Caco-2 cells [36,37]. Micelles are only useful for poorly water-soluble drugs; they undergo fast, diffusion-controlled release and have been reported to undergo dissociation following administration, releasing drugs prematurely [38]. Thus, a combination of liposomes and micelles offers the opportunity to increase drug loading and control release, gaining the benefits of each nanostructure [39,40].

It is clear that further innovation is required to better overcome the physical and biological challenges of cancer drug delivery using liposomes and micelles and to advance chemotherapy in the clinic [41,42,43]. Previously, we reported on the development of a liposome-micelle-hybrid (LMH) delivery system using the model insoluble drug lovastatin [39]. We now propose to employ the new LMH technology for the co-delivery of 5-FU and PTX with the aim of enhancing drug loading, controlled release (with potential thermo-responsiveness), cellular uptake in human cervical adenocarcinoma (HeLa) cells, and drug permeability.

## 2. Materials and Methods

### 2.1. Materials

1,2-Dipalmitoyl-sn-glycero-3-phosphorylglycerol (DPPG), 1,2-Distearoyl-sn-glycero-3-phospho-1’-rac-glycerol, sodium-salt (DSPG-Na) with C18:0, <99% (molecular weight: 801.058), and phosphatidylcholine (PC) with C18:0, <98% were purchased from Avanti polar lipids (Alabaster, AB, USA). PTX was purchased from Sinopharm Chemical Reagent Co. Ltd. (Shanghai, China). 5-FU was supplied by Beijing Mesochem Technology Co. (Beijing, China). d-α-Tocopheryl polyethylene glycol 1000 succinate (TPGS) was purchased from Antares Health Products, Inc. (Jonesborough, TN, USA).

Human cervical adenocarcinoma (HeLa) and Neuro 2A cells were purchased from the American Type Culture Collection (ATCC, Manassas, VA, USA), and Human epithelial colorectal adenocarcinoma cells (Caco-2) were obtained from the ATCC and kindly donated by the School of Medical Science, University of Sydney (Sydney, NSW, Australia). Dulbecco’s modified eagle’s medium (DMEM) with [+] 4.5 g/L-D glucose, [+] L-glutamine, sodium pyruvate, 1% nonessential amino acids, and 10% Fetal bovine serum (FBS), Trypsin 0.25% (*w*/*v*) in Phosphate-Buffered Solution (PBS), Cholesterol (CHO), Polyethylene glycol (PEG) 1500 and 400, dimethyl sulfoxide (DMSO), PBS tablets, HPLC grade acetonitrile, Propidium iodide, and Hoechst 33342 were purchased from Sigma Aldrich (St. Louis, MO, USA). Alamar Blue^®^ (Cat; DAL1025) was purchased from Thermo Fisher Scientific, Waltham, MA, USA.

Caco-2 studies were undertaken using 24-well polystyrene plates with inserts and lids (a polycarbonate Transwell filter with 0.4 μm pores and a surface area of 0.7 cm^2^). The transepithelial electrical resistance of the Caco-2 monolayers was determined using a Millicell ERS-2 voltohmmeter (Millipore Corporation Ltd., Bedford, MA, USA). Transport buffer Hank’s balanced salt solution (HBSS) was purchased from Gibco Life Technologies (Camarillo, CA, USA). A Class II cabinet with a laminar flow hood, fluorimeter, and monochromator plate reader (BioTek Instruments Inc., Winoski, VT, USA) was used and provided by the UNSW cell culture facility in the school of medical science (Lowy Children’s Cancer Center). All reagents: Sodium Chloride (NaCl), Sodium hydroxide (NaOH), Phosphoric acid (H_3_PO_4_), Tween 20, analytical grade chloroform, methanol, and ethanol were ordered from Ajax Chemicals (Scoresby, VIC, Australia). Ultrapure MilliQ water was used for all experiments and generated by a Milli-Q^®^ Ultrapure water system connected with a Q Gard^®^ purification cartridge and a Quantum^®^ EX polishing cartridge.

### 2.2. Preparation of PTX-Encapsulated Micelles

The method used for the preparation of PTX-loaded micelles was modified from the direct dissolution and solvent evaporation method described by Romana et al. [39] Briefly, different quantities of PTX (5, 10, and 15 mg) were mixed with 100 mg of TPGS and then dissolved in chloroform. The solvent was evaporated by a rotary evaporator for 2 h at 38 °C while rotating at 60 rpm. The film was then hydrated with a 10 mL PBS solution containing 0.9% sucrose. The hydrated solution was sonicated for 30 min to form micelles. A clear, drug-loaded micelle solution was formed. The free drug was separated by centrifugation at 10,000 rpm for 20 min.

### 2.3. Preparation of PTX-Encapsulated Thermosensitive Liposomes

PTX-loaded liposomes were prepared by combining DPPG, DSPG, PC, cholesterol, and PEG-1500 in a molar ratio of 80:5:5:2.5:7.5 by the thin-film hydration (TFH) method, according to Romana et al. [39] and Bangham et al. [44]. The lipids (100 mg) and PTX (10, 15, 20, and 25 mg) were dissolved in a mixture of methanol, chloroform, and water (10 mL) with a ratio of 1:5:0.2 in a round bottom flask by gentle handshaking and sonication (1–2 min continuous sonication at 20,000 Hz in direct mode) to form a clear solution. The excipient-solvent mixture was subjected to vacuum evaporation at 60 °C for 3 h (BUCHI rotavapor R-124 and BUCHI water bath B-480) until complete evaporation of the solvents produced a thin drug-lipid film. This process was above the phase transition temperature (Tc) of the lipids (55 °C). The rotation speed was kept at 60 rpm.

The homogenous thin lipid film was further dried for an hour with flowing N_2_ gas and kept under a vacuum in a hood overnight to remove the solvents completely. The resultant film was hydrated with 0.9% (*w*/*v*) sucrose in phosphate-buffered saline (PBS) for 2 h in a water bath (60 °C) with constant rotation at a slow speed. The hydrated lipid mixture was subsequently sonicated in an ice bath for 10 min. The drug-loaded liposomes were separated from the unencapsulated free drug by ultracentrifugation (32,000 rpm at 4 °C, Avanti JXN-30, Beckman Coulter Life Science, NSW, Australia). The collected pellets were suspended in the hydration media and extruded through polycarbonate filters (400–800 nm pore diameter) 8–10 times to obtain highly monodispersed (PDI < 0.25) and unilamellar liposomes. The final liposome product appeared as a clear, transparent, light blue-white suspension. The liposome suspension was stored in refrigerator conditions (4 °C) until used.

### 2.4. Preparation of PTX-Encapsulated Thermosensitive Liposome-Micelle Hybrid (LMH_PTX_)

To fabricate the LMH systems, PTX-loaded TPGS micelles were used in the hydration step of liposomes in the TFH method described in Section 2.3. After rehydration, the suspension was sonicated and ultracentrifuged in the same conditions reported for PTX-loaded liposomes. PTX present in the pellet (encapsulated within LMH) and supernatant (unencapsulated drug) was analyzed separately. The LMH pellets were re-suspended in hydration solution and extruded, as per the liposome preparation method described in Section 2.3. LMH were previously reported to be stable for more than 1 month upon storage in the fridge [39].

### 2.5. Preparation of PTX-5-FU-Loaded LMH_PTX-5-FU_

Firstly, 5-FU (10, 15, 20, and 25 mg) was mixed with 100 mg of TPGS for the 5-FU micelles formulation. The micelle preparation method then followed the same procedure described in Section 2.2 for PTX-loaded micelles.

To prepare 5-FU-loaded LMH, 5-FU-loaded micelles were used to hydrate the PTX-loaded liposome film for LMH_PTX-5-FU_ prepared as per Section 2.3. After preparation of LMH_PTX-5-FU_, sonication-ultracentrifugation-extrusion methods followed the same procedures as described in Section 2.4. 5-FU-loaded liposomes were not prepared because 5-FU was loaded in the LMH core only and was not suitable for loading in the phospholipid bilayer due to its hydrophilic nature.

Blank micelles, liposomes, and LMH were also prepared. All samples were freeze-dried with sucrose as a cryoprotectant (Martin Christ Freeze Dryer, D-37520) at −50 °C and 0.001 bar in preparation for cell culture studies.

### 2.6. Encapsulation Efficiency and Drug Loading

PTX and 5-FU nanocarrier formulations were prepared by diluting 100 μL of each formulation with 900 μL of acetonitrile and vortexed to disrupt the carriers. The samples were centrifuged at 10,000 rpm (5000× *g*) for 10 min to separate the filtrate and filtered through 0.22 μm PTFE syringe filters. The concentrations of PTX and 5-FU were analyzed by HPLC (described below). The encapsulation efficiency (EE%) was calculated as a percentage ratio between the quantity of drug encapsulated in the nanocarriers and the initial drug added, and the drug loading (DL%) was expressed as the amount of drug entrapped in the nanocarriers compared to the total nanocarrier weight.

### 2.7. PTX Assay Method

The concentration of PTX was analyzed using HPLC (Shimadzu UFLC Model LC-20 AD) with an X-Bridge™ C18 column (156 × 10.0 mm), Waters Corporation, Milford, MA, USA. The mobile phase was a mixture of 45% (*v*/*v*) of acetonitrile and 55% (*v*/*v*) of water. An isocratic elution method was used. The flow rate was set at 1.0 mL/min with a run time of 10 min, and the absorbance was measured at 227 nm. Samples were injected at a volume of 50 µL at room temperature. A series of working solutions with known concentrations were used to generate a linear calibration curve (n = 4) by plotting the chromatographic peak area versus PTX concentration.

### 2.8. 5-FU Assay Method

5-FU concentration was analyzed using the same HPLC and column as above, with a mobile phase of methanol (10% (*v*/*v*)) and PBS (90% (*v*/*v*) PBS). The mobile phase was degassed via ultrasonication for 30 min before use. The absorbance was measured at 254 nm. Samples (50 µL) were injected at room temperature. A linear calibration curve (n = 4) was generated using a series of working solutions by plotting the chromatographic peak area versus 5-FU concentration.

### 2.9. Characterization of Liposome-Micelle-Hybrid (LMH) Nanocarriers

#### 2.9.1. Particle Diameter and Size Distribution

Dynamic Light Scattering (DLS) and Phase Analysis Light Scattering (PALS) (Zetasizer Nano ZSP (Malvern Panalytical, Malvern, UK)) were used to determine the average particle diameter (z-average), size distribution (polydispersity index, PDI), and zeta potential of the nanocarriers. The micelles were analyzed without dilution, while liposomes and LMH were diluted 100 times in Milli-Q water prior to analysis. Zeta potential was measured in PBS (10^−3^ M). Each sample was measured three times at 25 °C, and the material RI was 1.59.

#### 2.9.2. Differential Scanning Calorimetry

The lipid phase transition temperature was assessed by DSC (TA Instruments, New Castle, DE, USA) for all drug-loaded nanocarriers. DSC measurements were performed by employing nitrogen flow (50 mL/min) using a heating rate of 2 °C/min, an empty pan as a reference. Two heating/cooling scans are carried out from 20 °C to 70 °C, and the transition temperature, *T*_m_ as well as the temperature width at half maximum of the DSC were determined by Prism^®^ software version 8 (GraphPad, San Diego, CA, USA).

#### 2.9.3. Morphological Characterization of Nanocarriers

Transmission electron microscopy (TEM) was used to characterize the micelles, liposomes, and LMH morphology. Samples were first diluted (PBS) prior to a drop of sample (6 µL) being applied on a Formvar-coated copper grid (200 mesh size) for 1 min, and excess was removed with filter paper. Samples were subsequently negatively stained with uranyl acetate (2%, 20 µL) for 60 s, and the excess stain was blotted away with filter paper. The grid was then dried overnight in air, and TEM micrographs were recorded on an FEI Tecnai G2 20 (Eindhoven, Netherlands) from the Electron Microscope Unit at the Mark Wainwright Analytical Centre of the University of New South Wales (Sydney, NSW, Australia).

### 2.10. Temperature-Triggered Release of PTX and 5-FU from Nanocarriers

In vitro drug release from the nanocarriers was studied in PBS buffer (pH = 7.4) in the presence of 0.5% PEG-400 as a solubilizer to maintain sink conditions in the release medium. Each formulation (4 mL) was tightly sealed in a dialysis bag (MWCO 12 kDa, Sigma Aldrich, St. Louis, MO, USA) and immersed in 40 mL of release medium. A MWCO of 12 kDa was selected to ensure drug transport through the membrane was not a contributing factor. The release study was performed at 37 °C and 42 °C, and mixing was achieved using a magnetic stirrer (100 rpm). At defined time intervals, an aliquot of 0.5 mL was taken from the release medium, followed by an immediate replacement with an equal volume of the fresh media. The samples were dissolved in equal volumes of acetonitrile, centrifuged at 10,000 rpm (5000× *g*) for 10 min to separate the supernatant, and analyzed by HPLC.

### 2.11. In Vitro Drug Release Kinetics

Drug release data for PTX and 5-FU from the micelles, liposomes, and LMH (at 37 °C and 42 °C) were fitted with the Korsmeyer-Peppas [45] model as described in Equation (1).
*M_t_*/*M_∞_* = *Kt^n^*(1)

This model has previously been used to describe drug release from polymeric systems where *M_t_*/*M_∞_* is the fractional drug release (usually expressed as %), *K* is a characteristic kinetic constant that depends on the rate of degradation and dissolution, and *n* is an exponent coefficient that characterizes the mechanism of release (either diffusion, swelling/relaxation, or a combination of both).

To elucidate more details of the release mechanism, the Korsmeyer-Peppas model can be extended to incorporate the diffusion coefficient (*D*) for the drug molecule in the nanocarrier matrix [39], when *n* = ½, as shown in Equation (2).
*M_t_/M_∞_* = 4(*D_t_/πλ*^2^)^1/2^
(2)

where *D_t_* is the diffusion coefficient at time *t* and *λ* is the thickness of the nanocarriers.

### 2.12. Permeability Assessment

Permeability assessment of PTX-loaded micelles, liposomes, and LMH followed the same procedure as described in Romana et al. [39]. Briefly, to prepare a Caco-2 monolayer, cells were seeded at a density of approximately 40,000 cells/cm^2^ on polycarbonate Transwell filters (0.4 μm pores and a surface area of 0.7 cm^2^) in 24-well polystyrene plates and maintained in an incubator in DMEM for 21 days. The medium was changed on alternate days, first the basolateral (400 μL) and then the apical (600 µL). At the end of days 14 and 21, transepithelial electrical resistance (TEER) was used to assess the integrity of the monolayers in the culture medium using a Millicell ERS-2 voltohmmeter.

TEER values (Ω·cm^2^) were calculated by subtracting the resistance of the blank media (DMEM + FBS-10%) without cells from the total resistance and then multiplying by the effective membrane area (0.49 cm^2^). The control TEER value was <200 Ω·cm^2^ and remained constant for the duration of the experiment. The average TEER value in Caco-2 cell monolayers (CCM) containing media was found to be 1036 ± 327 Ω·cm^2^ on day 14 and 1363 ± 262 Ω·cm^2^ on day 21 of culture. These indicated a complete cell monolayer had been developed by 14 days. All TEER values were above 305 Ω·cm^2^, indicating the integrity of the cell monolayer was maintained [46].

For this study, PTX-loaded nanocarriers were dissolved in 0.5% (*v*/*v*) of DMSO and HBSS to prepare experimental samples (LMH, along with individual samples of liposomes, micelles, and free PTX for comparison). Each of the nanocarriers and free PTX samples contained 100 μM of PTX. After 21 days, the DMEM (growth medium) from the apical chamber was replaced by the transport buffer HBSS for both control and sample (400 μL in the apical wells and 600 μL in the basal wells) for 30 min. Subsequently, HBSS buffer was replaced by samples in the apical (400 μL) or basal (600 μL) wells. After an incubation of 2 h, a sample (500 μL) was removed from the appropriate well to calculate an efflux ratio (ER) as described in Equation (4).

After treatment, cells were washed three times with PBS. The solution in the basolateral chamber was collected and lysed with methanol. Samples were quantitatively analyzed for PTX by HPLC as described above.

The apparent permeability (*P_app_*) and efflux ratio were determined from Equations (3) and (4) [47].
a.The apparent PTX permeability (*P_app_* × 10^−6^ µg/s) was calculated as follows:
(3)$$P_{\mathrm{app}}=\frac{V_{R}}{A\times C_{0}\;}\times \frac{dM_{t}}{dt}$$

*V_R_* is the volume of the receiving chamber; *A* = monolayer filter area (cm^2^); *C_o_* = mass of the compound initially in the donor compartment; and *dM_t_*/*dt* = the rate of drug permeation across the cells.
b.The efflux ratio (ER) was calculated as the ratio of *P_app_* determined in the A-to-B direction to *P_app_* determined in the B-to-A direction:
ER = (*P_app_* B–A)/(*P_app_* A–B)(4)
where the ratio of the basolateral-apical (secretion) component *P_app_* B–A to the apical-basolateral (absorption) component *P_app_* A–B was assessed. Theoretically, an ER superior to unity implies the action of one or various efflux transporters on the tested compound.

### 2.13. Cell Viability Studies

HeLa and Neuro 2a Cells were grown in DMEM supplemented with 10% (*v*/*v*) fetal bovine serum (FBS) and glutamine (1%). The cells were maintained in an incubator supplied with 5% CO_2_ and a 95% air-humidified atmosphere at 37 °C. Suspension samples of cells in growth media were seeded into 96-well tissue culture plates at a density of 1 × 10^4^ cells per well and allowed to attach overnight.

Initially, the nanocarrier samples were prepared by dissolving them in DMEM and FBS (10%), and 0.5% (*v*/*v*) of DMSO was used to dissolve PTX + 5-FU. A total of nine concentrations of each formulation were prepared and placed in 96-well cell plates for 48 h at both 37 °C (CellXpert^®^ C170-Cell Culture Incubator, Eppendorp, Hamburg, Germany) and 42 °C (HERACELL VIOS 160i CO_2_ incubator, ThermoFisher, Waltham, MA, USA). After 48 h, the media was aspirated, and Alamar Blue solution was directly added to the medium, resulting in a final concentration of 10% in each well. After 3.4 h of incubation with Alamar Blue^®^, absorbance was measured at 570 and 596 nm using a BioRad microplate reader (ELX800, Biotek, Santa Clara, CA, USA). Untreated cells were taken as controls with 100% viability, and cells with the addition of drug-free nanocarriers were used as blanks. Results were expressed as cell viability (%) as an absorbance ratio between cells treated with free drug or drug-loaded formulations and the absorbance of cells without any drug treatment.

Cellular morphology analysis was performed by double staining with Hoechst 33342 and propidium iodide. After incubation of the selected samples (5-FU+PTX, LMH) in cells in a 96-well plate, the cells were stained with 0.5 μg/mL Hoechst 33342 and 1 μg/mL propidium iodide and left for 10 min. The cells were then immediately imaged using an Olympus CellR epifluorescence microscope (XM10, Olympus, Japan). Microscopy images were used to highlight the difference between living and dead cells. Here, Hoechst acts as a marker for all cells, while propidium iodide is selective for apoptotic or dead cells only. The microscopy data was processed with ImageJ software.

### 2.14. Statistics Analysis Method

Statistical analysis was performed using one-way analysis of variance (ANOVA) using the statistical software package SPSS. The data were expressed as mean ± standard deviation (SD). A multiple range test was used to compare each group, and the resulting *p* values are indicated in the figures.

## 3. Results and Discussion

### 3.1. Preparation and Characterization of Liposome, Micelle, and LMH Formulations

PTX-loaded liposomes and PTX- and 5-FU-loaded micelles were initially prepared. PTX-loaded micelles were encapsulated into the core of PTX-loaded liposomes to form double-loaded PTX-LMH (LMH_PTX-PTX_); 5-FU-loaded micelles were encapsulated into PTX-loaded liposomes to prepare LMH_PTX-5-FU_; and PTX-loaded micelles were incorporated into the liposomes to form LMH_PTX_. The size and zeta potentials of the nanocarriers are given in Table 1.

In line with expectations, the micelles (loaded and unloaded) displayed mean diameters in the range 10–13 nm and the liposomes and LMH in the range 150–175 nm with PDI < 0.3. A trend of a small increase in liposome size was observed upon incorporation of the drug and drug-loaded micelles; this is consistent with previous results [39,48]. The measured zeta potentials of liposomes and LMH were in the range −29 to −35 mV, which is as expected for DSPG and DPPG lipid-based systems and of an appropriate magnitude for good colloidal stability [49,50]. Transmission electron microscopy (TEM) images of micelles, liposomes, and LMH systems are provided in Figure 1.

The micelles (Figure 1a) were revealed to be roughly spherical with sizes in the 10–20 nm range, in agreement with DLS analysis. The liposomes (Figure 1b) and LMH (Figure 1c) images confirmed sphericity, with some unilamellar character for liposomes and high contrast for the LMH, which contained encapsulated micelles and higher drug loading. Previous studies on chitosan-coated curcumin nanoliposomes and PTX-loaded PEGylated liposomes have shown similar features [51]. The size of the nanocarriers (<180 nm) is appropriate for effective delivery, with potential for enhancing circulation time and tumor delivery [52].

Differential scanning calorimetry (DSC) data for both drug-loaded and blank nanocarriers is given in Figure 2 and identifies differences in thermal transitions. TPGS micelles showed the lowest (and sharpest) transition temperature at 34.5 °C, which was independent of drug loading. In general, a higher phase transition temperature was observed for liposomes and LMH, which is considered due to the combination of cholesterol and phospholipids, the lamellar structure, and thermal stability [53,54]. More specifically, blank liposomes showed a broad thermogram with a phase transition temperature around 39 °C; this sharpened and shifted to ~45.5 °C with PTX loading. This is in agreement with reports that PTX causes interdigitation and the formation of a stable gel phase [55]. The blank LMH and drug-loaded LMH showed phase transition temperatures of ~41 and 39.5 °C, respectively. The lower phase transition temperature (*T_m_*) for the LMH_PTX-5-FU_ compared to the drug-loaded liposomes is linked to the crystalline to liquid phase behavior of the DPPG bilayer at 40–42 °C (the phase transition of DPPG is 41.5 °C [56]), which may facilitate temperature-dependent drug release, which is carrier type-dependent; this is explored in subsequent sections.

### 3.2. Drug Loading Behaviour

Drug loading data for each nanocarrier is reported in Table 2. PTX loading in the liposomes (4.72 ± 0.40%) was higher than for the TPGS micelles (2.20 ± 0.14%) and LMH_PTX-5-FU_ (3.12 ± 0.53%). Importantly, the PTX loading increased in LMH_PTX-PTX_ (6.04 ± 0.13%) due to the combination of the two loading environments (Table 2). PTX loading (1.42 ± 0.03%) within the micelle core of LMH_PTX_ was lower than when directly loaded in micelles (2.20 ± 0.14%). Similarly, 5-FU loaded in LMH_PTX-5-FU_ was slightly lower than directly in micelles. Importantly, the dual-loaded LMH_PTX-PTX_ enabled PTX to be dosed at a 208-fold increased solubility compared with the pure drug.

### 3.3. Temperature-Dependent Drug Release from Micelles, Liposomes, and LMH

In vitro release of PTX and 5-FU from LMH, liposomes, and micelles at physiological (37 °C) and hyperthermia (42 °C) temperatures in sink conditions is presented in Figure 3.

PTX release kinetics from the standalone micelle and liposome formulations are rapid, which can be undesirable and problematic when considering premature release and the potential for precipitation of poorly soluble drugs during application. Ji et al. [57] reported similar PTX release from liposomes. Significantly, the LMH carriers provide considerably slower and sustained release; this has potential advantages when considering drug delivery applications, e.g., crossing biological barriers, sustained circulation, and passive targeting. This controlled release behavior is considered due to the combined barriers of micelles and lipid bilayers, which provide synergy in reducing transport from the nanocarrier into the external aqueous environment. 5-FU is released quickly from the micelles, and the sustained release of 5-FU from LMH is less pronounced and reflects the drug’s lower molecular weight and higher water solubility in comparison to PTX.

It is also apparent that drug release is more pronounced at 42 °C than at 37 °C and that the temperature dependence is carrier type dependent, with the LMH formulation showing a great temperature dependence. The observed increase in drug release at 42 °C compared to 37 °C is due to at least two potential mechanisms. Firstly, due to increased drug diffusion, and secondly, due to the transition temperature of the lipids in the formulation, i.e., increased barrier transport kinetics at temperatures above the transition temperature. For example, the transition temperature of the DPPG liposome system used here is exceeded as the temperature increases from 37 to 42 °C, triggering a change in the lipid bilayer packing from an ordered gel phase to the disordered liquid crystalline phase, which reduces the barrier for drug release [27].

In more quantitative terms, after 48 h at 37 °C, PTX release from liposomes, LMH_PTX_, and LMH_PTX-5-FU_ was ~87%, 50%, and 27%, respectively. LMH_PTX_ showed a clearly different drug release profile compared with liposomes, which may be a result of TPGS molecules from the micelles influencing the structure of the lipid bilayer as reported for other non-ionic surfactants when interacting with liposomes [57]. The highly sustained release of PTX from LMH_PTX-5-FU_ may be due to 5-FU’s influence on the packing of the lipid bilayer, as observed previously for resveratrol release from dual-loaded 5-FU and resveratrol liposomes [30]. For 5-FU in LMH_PTX-5-FU_, 60% release occurred in 1 h at 37°C, which increased to 84% in 1 h and 100% in 2 h at 42 °C. Clearly, 5-FU can readily cross both the micelle and lipid bilayer barriers [22].

### 3.4. Analysis of Release Kinetics

Following a previously reported approach, the Korsmeyer and Peppas kinetic model was used to better understand the PTX release behavior and mechanism [45]. *K*-t plots obtained using Equations (1) and (2) are shown in Figure 4, with the associated *n* and *D* values given in Table 3. Since *n* is observed to be <0.5 for PTX release from all nanocarriers at both temperatures, Fickian diffusion via a potential chemical gradient is most likely [58].

The calculated rate constant (*K*) and diffusion coefficient (*D*) for PTX gradually increased in the order of LMH_PTX-5-FU_ < LMH_PTX_ < micelles < liposomes (Table 3). The fitted curves are shown in Figure 4 (dotted and dashed lines). The rate constant and the diffusion coefficient are the lowest (0.14 × 10^−10^ m^2^/s) for the LMH_PTX-5-FU_ system, which suggests the slowest diffusion of the drug molecules. For other nanocarriers, a trend of increased diffusion coefficients of 0.70 × 10^−10^, 1.53 × 10^−10^, and 2.49 × 10^−10^ m^2^/s was observed for LMH_PTX_, micelles, and liposomes, respectively. This is in agreement with our previous work, where the rate of drug diffusion was limited by the liposomal lipid layer around the micelles in the LMH [39]. Rate constants and diffusion coefficients were greater at 42 °C in comparison to 37 °C; this correlated with the release profiles and is considered due to changes to the lipid-bilayer ordering and explained by classical kinetic theory [59].

### 3.5. Permeability Behavior of LMH Nanocarriers

The influence of nanocarrier type on PTX permeability was investigated in Caco-2 cell monolayers following our previous LMH studies [39]. For free and nanocarrier-encapsulated PTX, the TEER values decreased (Figure 5a) by 63%, 60%, 55%, and 31% after 2 h for micelles, liposomes, LMH_PTX_, and free PTX, respectively. A similar TEER reduction was reported previously for LOV-LMH [39], liposomes [60], and micelles [61]. The increased transport is considered due to the opening of the Caco-2 cell monolayer tight junctions and is correlated with the observed increase in the PTX permeability coefficient (Figure 5b) [62]. The apparent permeability coefficient of PTX was significantly increased compared to unformulated PTX when using all the nanocarriers.

### 3.6. In Vitro Cell Viability Studies

HeLa and Neuro 2a (N2a) are cancer-derived cells, and we have determined their viability (at both 37 °C and 42 °C) against PTX concentration (free PTX and encapsulated in micelles, liposomes, and LMH_PTX-5-FU)_ (Figure 6). The free carriers, TPGS micelles and DPPG liposomes, have been investigated for delivery of many chemotherapeutics and shown to be well tolerated by cells (>80% viability) at the concentrations used in this study [63,64]. 5-FU-loaded micelles were not evaluated in cellular studies due to their fast drug release profile, making them unsuitable for testing.

A key observation is that the LMH_(PTX-5-FU)_ formulation is significantly more cytotoxic against both cell lines (and at both temperatures) than the free drug combination (PTX+5-FU), PTX-loaded micelles, and PTX-loaded liposomes. At 37 °C, the LC50 values for LMH_PTX-5-FU_ are ~1.0 and ~0.5 μM against Hela and Neuro 2a, respectively; these are ~10 times lower than for the other formulations and free drugs. Furthermore, LC50 values for LMH_PTX-5-FU_ are significantly reduced (more than five times lower) at 42 °C compared with 37 °C. The high cytotoxicity of LMH_PTX-5-FU_ concurs with high drug loading levels and sustained drug release. The observed change in cell viability when PTX and 5-FU were delivered using LMH may be hypothesized to be due to increased uptake of the nanocarrier into the cell, resulting in cell death.

This behavior is confirmed by the cellular images and cell morphology data in Figure 7. That is, a significantly higher number of cells appear damaged and have undergone apoptosis for LMH_PTX-5-FU_ treatment compared with free PTX+5-FU and in both 37 °C and 42 °C (Figure 7A,B), which is supported by the significantly higher number of dead cells for the LMH_PTX-5-FU_ treatment (Figure 7C,D), which is more pronounced at the higher temperature. It can be hypothesized that this behavior is also related to the improved cellular delivery of PTX and 5-FU using the LMH nanocarrier.

Overall, these results revealed that LMH_PTX-5-FU_ (PTX- and 5-FU-loaded LMH nanocarriers) are more effective as anti-cancer agents compared with micelles, liposomes, and free PTX+5-FU, with strong thermo-responsive characteristics.

## 4. Conclusions

The LMH system for PTX and 5-FU was successfully developed with optimum particle size distribution and increased drug loading compared with single micelle or liposome formulations. The LMH nanocarrier facilitated sustained PTX release (not 5-FU) and increased the permeability and transport of PTX across Caco-2 cell monolayers in comparison to unformulated PTX. PTX-loaded LMH significantly enhanced cytotoxicity against HeLa and Neuro 2a cells with extensive temperature dependency (cytotoxicity at 42 °C > at 37 °C). LMH nanocarriers offer potential as a next-generation cancer drug delivery system with advantages over conventional liposomes and micelles for improving the therapeutic efficacy of anticancer drugs for future exploitation in cancer medicine.

## Figures and Tables

**Figure 1 pharmaceutics-15-01886-f001:**
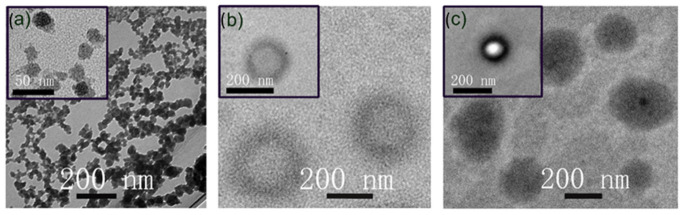
Representative TEM images of the nanocarriers: (**a**) micelles, (**b**) liposomes, and (**c**) LMH carriers.

**Figure 2 pharmaceutics-15-01886-f002:**
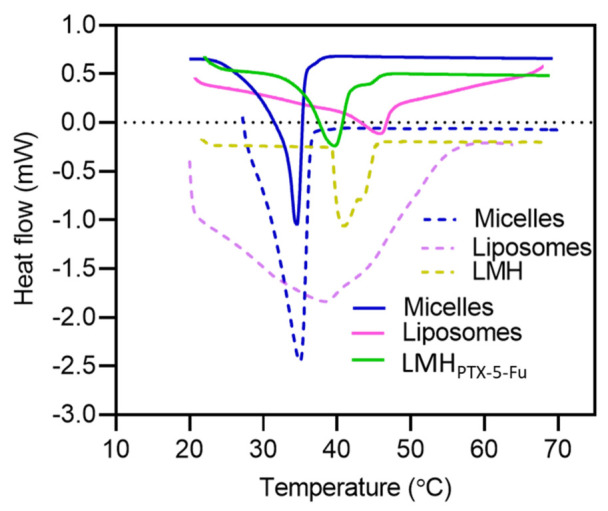
DSC profile of micelles, liposomes, and LMH_PTX-5-FU_. The heating scan rate for all nanoparticles was 2 °C min^−1^ from 20 to 70 °C. The dotted lines represent blank micelles, liposomes, and LMH, respectively.

**Figure 3 pharmaceutics-15-01886-f003:**
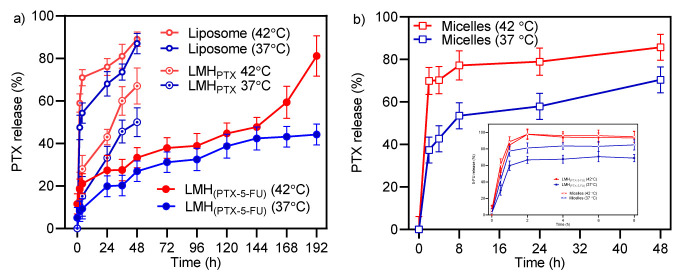
(**a**) Temperature-dependent PTX release from liposomes, LMH_PTX_ and LMH_PTX-5-FU_. (**b**) Temperature-dependent PTX release from micelles and 5-FU release from micelles and LMH_(PTX-5-FU)_ as inset. (mean ± SD (n = 3)).

**Figure 4 pharmaceutics-15-01886-f004:**
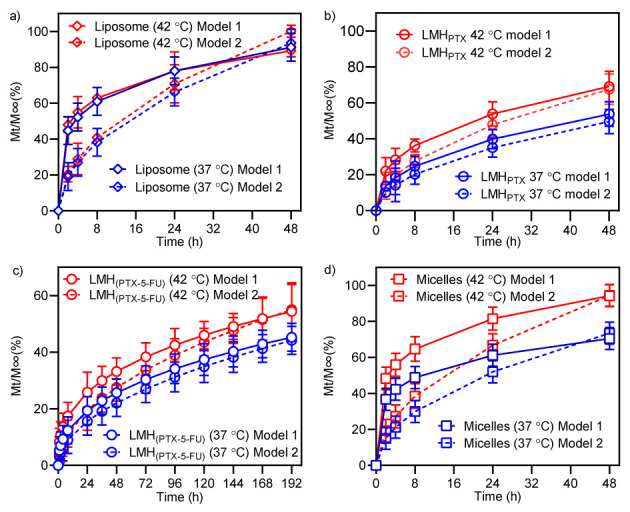
Kinetics of drug release data fitted with models 1 and 2 for (**a**) LMH_PTX-5-FU_, (**b**) micelles, (**c**) liposomes, and (**d**) LMH_PTX_ The solid line is model 1, and the dotted line area is model 2. The blue data represents 37 °C, and the red indicates 42 °C.

**Figure 5 pharmaceutics-15-01886-f005:**
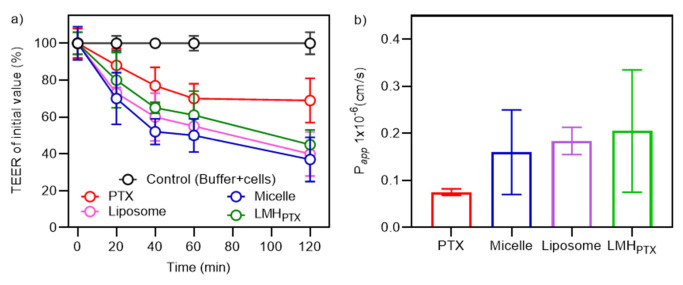
(**a**) TEER of Caco-2 cell monolayers (21 days old) in HBSS after PTX (100 µM) treatment (free and in micelles, liposomes, and LMH_PTX_ formulations). (**b**) Apparent permeability coefficient for PTX.

**Figure 6 pharmaceutics-15-01886-f006:**
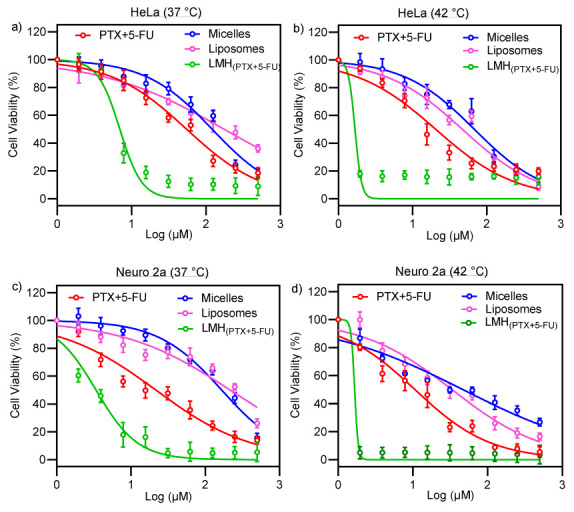
Viability of HeLa (**a**) at 37 °C and (**b**) at 42 °C and Neuro 2a (**c**) at 37 °C and (**d**) at 42 °C cells after 48 h of treatment against PTX-loaded micelles (blue circles), PTX-loaded liposomes (pink circles), PTX- and 5-FU-loaded LMH_PTX-5FU_ (green circles), and the free drug combination of PTX and 5-FU (PTX+5-FU; red circles) (mean ± SD, n = 4).

**Figure 7 pharmaceutics-15-01886-f007:**
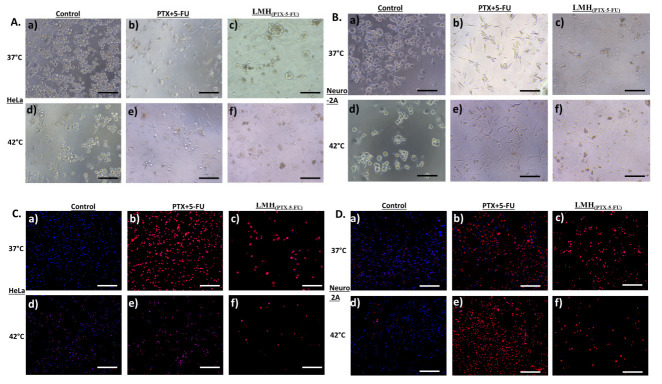
Bright field microscopic images of (**A**) Hela (upper left) and (**B**) Neuro 2a (upper right) cells: control and PTX+5-FU and LMH_PTX-5-FU_ treatments at 37 °C ((**a**), (**b**) and (**c**), respectively) and 42 °C ((**d**), (**e**) and (**f**), respectively) (scale bar = 1000 µm). With corresponding fluorescence microscopy images of (**C**) HeLa (bottom left) and (**D**) Neuro (bottom right) (scale bar = 500 µm): control and PTX+5-FU and LMH_PTX-5-FU_ treatments at 37 °C ((**a**), (**b**) and (**c**), respectively) and 42 °C ((**d**), (**e**) and (**f**), respectively) N.B. cells treated with propodium iodide and Hoechst 33342: live cells are blue and dead cells are red.

**Table 1 pharmaceutics-15-01886-t001:** Particle diameter, polydispersity index (PDI), and zeta potential of micelles, liposomes, and LMH in the presence or absence of PTX or 5-FU loading (mean ± SD, n = 3).

Nanocarrier Type		Mean Diameter (nm)	PDI	Zeta Potential (mV)
Micelle	Blank	13 ± 0.4	0.242 ± 0.027	
	5-FU-Loaded	12 ± 0.3	0.059 ± 0.005	
	PTX-Loaded	10 ± 0.1	0.039 ± 0.007	
Liposomes	Blank	154 ± 2.3	0.121 ± 0.015	−32.8 ± 0.4
	PTX-Loaded	167 ± 4.5	0.276 ± 0.024	−31.1 ± 0.1
LMH	Blank	151 ± 2.6	0.087 ± 0.021	−34.1 ± 0.2
	LMH_PTX_	157 ± 2.5	0.256 ± 0.029	−29.6 ± 0.7
	LMH_PTX-5-FU_	164 ± 1.9	0.171 ± 0.028	−32.9 ± 0.6
	LMH_PTX-PTX_	175 ± 1.7	0.217 ± 0.0208	−30.0 ± 0.7

**Table 2 pharmaceutics-15-01886-t002:** PTX and 5-FU loading in micelle, liposome, and LMH nanocarriers (mean ± SD, n = 3).

Nanocarrier Type (Loaded Phase)	Drug Loading (mg/mL)		Aqueous Solubility Improvement (Fold Increase)
	PTX	5-FU	PTX *
Micelle (core)	2.20 ± 0.14	3.16 ± 0.36	73.4
Liposome (bilayer)	4.72 ± 0.40	-	157.3
LMH_PTX_ (core)	1.42 ± 0.03	-	47.3
LMH_PTX-5-FU_ (PTX bilayer and 5-FU core)	3.12 ± 0.53	2.91 ± 0.41	103.3
LMH_PTX-PTX_ (bilayer and micelles in the core)	6.04 ± 0.13	-	208.6

* Solubility of PTX in water = 0.03 mg/mL.

**Table 3 pharmaceutics-15-01886-t003:** The *K*, *n*, and *D* were determined from the Korsmeyer and Peppas model, Equations (1) and (2), using non-linear fitting (MATLAB-R2017a) and the Microsoft Excel solver as a function of time over 192 and 48 h. R^2^ values were >0.9 for LMH_PTX_ (both models), LMH_PTX-5-FU_, liposomes (model 2), and micelles (model 2), and >0.75 for liposomes (model 1) and micelles (model 1).

Nanocarrier Type		37 °C			42 °C	
	*K*	*n*	*D* (m^2^/s) ×10^−10^	*K*	*n*	*D* (m^2^/s) ×10^−10^
LMH_PTX-5-FU_	0.187	0.408	0.138	0.447	0.357	0.215
LMH_PTX_	0.290	0.433	0.699	0.900	0.361	1.301
Liposome	6.117	0.204	2.491	8.501	0.195	2.850
Micelle	5.957	0.205	1.532	7.447	0.214	2.532

## Data Availability

The datasets generated and/or analysed during the current study are available from the first author on reasonable request.

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
