# Peer review of "Liposome-Micelle-Hybrid (LMH) Carriers for Controlled Co-Delivery of 5-FU and Paclitaxel as Chemotherapeutics"

_pharmaceutics, 2023, doi:10.3390/pharmaceutics15071886_

Round 1

Reviewer 1 Report

The manuscript provides a comprehensive overview of the challenges associated with the clinical use of paclitaxel (PTX) and 5-fluorouracil (5-FU) in cancer chemotherapy. The section effectively highlights the limitations of PTX and 5-FU, such as poor oral bioavailability, low solubility, limited intestinal permeability, and the development of multidrug resistance (MDR) in cancer cells. While the authors have made significant efforts in investigating the development of a LMH system for PTX and 5-FU, there are several areas that require further attention and improvement. In this letter, I aim to highlight the key critiques in each section of the article.  

Comments for reviewer

Abstract:

1.      The opening sentence sets the stage by introducing the problem of biopharmaceutical challenges faced by PTX and 5-FU. However, it would be helpful to provide more specific examples of these challenges (such as poor solubility, low permeability, or lack of controlled release) to give the readers a clearer understanding of the issues being addressed.

2.      The statement that these challenges "reduce their effectiveness in new medicines" could benefit from further explanation or supporting evidence. It would be valuable to mention specific limitations or shortcomings of the current formulations and how they hinder clinical outcomes.

3.      Conclude by briefly explaining the specific attributes or advantages of LMH carriers that make them promise for cancer drug delivery and as next-generation.

Introduction:

1.      Provide specific citations for the statements regarding the limitations of PTX and 5-FU, such as low oral bioavailability, inadequate dissolution kinetics, solubility, poor intestinal permeability, and the development of MDR in cancer cells.

2.      Expand on the advantages of liposomes and micelles in overcoming the challenges associated with PTX and 5-FU, such as their ability to encapsulate insoluble drugs, improve drug stability, enhance drug circulation, and target specific cellular pathways.

3.      Briefly explain how the mentioned examples of nanomedicine approaches (liposomal doxorubicin, thermosensitive liposomes, PEGylated liposomes, and micelle-based formulations) address specific limitations of PTX and 5-FU, demonstrating their relevance to the current study.

4.      Highlight specific areas where innovation is needed in cancer drug delivery using liposomes and micelles, and briefly discuss the potential benefits that further innovation could bring.

5.      Lack of clarity in the research question or objective.

Methods:

1.      The summary provided earlier lacks specific details about the concentrations, ratios, and volumes used for the various materials and reagents. Providing more precise information would enhance the reproducibility of the study.

2.      The section does not mention the specific sonication parameters such as time, amplitude, or mode used during the preparation of liposomes and hybrid nanocarriers. Including these details would help readers understand the manufacturing process better.

3.      It would be helpful to include information on the stability and storage conditions of the prepared nanocarriers. This information is crucial for ensuring the integrity and functionality of the formulations during storage and transportation.

4.      The summary does not provide information about the cell lines used for in vitro experiments, cell culture conditions, or cell viability assays.

5.      The summary does not mention any in vivo experiments or animal models used to evaluate the efficacy or safety of the nanocarriers. If such experiments were conducted, it would be important to include details about the animal models, treatment protocols, and ethical considerations.

6.      While the summary mentions that the release data were fitted to the Korsmeyer-Peppas model, it does not provide any information on the validation or goodness-of-fit analysis of the model. Including this information would help assess the reliability and accuracy of the model in describing the release kinetics.

Results:

1.      The results section provides the experimental findings, but there is a lack of in-depth analysis and interpretation of the data. The authors could have discussed the implications of the particle size, zeta potential, and thermal transitions on the stability and functionality of the nanocarriers. Additionally, a more thorough discussion of the drug loading behavior and its implications for drug delivery could have been included.

2.      while the authors briefly mention some previous studies, there is a missed opportunity to compare the findings with existing literature. A more comprehensive discussion that highlights similarities and differences between this study and previous research on similar nanocarriers would provide a better context for the readers.

3.      The authors mention that drug release from the nanocarriers is more pronounced at 42°C compared to 37°C, but there is limited explanation or discussion on the underlying mechanism. It would be beneficial to delve deeper into the factors influencing temperature-dependent release and how it relates to the specific characteristics of the nanocarriers.

4.      The authors mention the use of the Korsmeyer and Peppas kinetic model to understand the drug release behavior, but there is minimal discussion on the model itself and how well it fits the experimental data. The authors could have provided more insight into the implications of the obtained K and n values and discussed the limitations of the model in capturing the release mechanism accurately.

Discussion

1.      According to the conclusion, the LMH nanocarrier demonstrated sustained release of PTX (but not 5-FU) and improved permeability and transport of PTX across Caco-2 cell monolayers when compared to micelle or liposome formulations. This finding suggests that the LMH nanocarrier could potentially enhance the effectiveness of PTX in cancer treatment. This needs some discussion.

Conclusions:

1.      the conclusion states that PTX loaded in LMH nanocarriers exhibited significantly enhanced cytotoxicity against HeLa and Neuro 2a cells. Notably, the cytotoxicity was found to be more pronounced at an elevated temperature of 42 °C compared to 37 °C, indicating a temperature dependency in the cytotoxicity of PTX loaded in LMH nanocarriers. This needs to be clearly stated and included in the discussion.

2.      LMH nanocarriers hold promise as a next-generation cancer drug delivery system with advantages over conventional liposomes and micelles. It is mentioned that LMH nanocarriers have the potential to improve the therapeutic efficacy of anticancer drugs, making them a viable option for future exploration in cancer medicine.

 Language and Style:

he writing in the provided sections is clear and concise. The language used effectively communicates the key points and findings of the study. The authors have presented their research in a structured manner.

Reviewer 2 Report

In this work, the authors prepared a liposome-micelle-hybrid (LMH) carriers for enhancement the co-formulation and delivery of PTX and 5-FU. The results are interesting, and might be helpful for the development of next generation chemotherapeutics. However, before the consideration of the publication, improvements are still necessary.

1. when the author do the temperature-triggered release assay, please explain why you choose 0.5 % PEG-400 as a solubilizer. And why you choose a dialysis bag with a relatively larger molecular weight (12 kDa), cause the drug PTX (Mw 853.92) and 5-FU (Mw 262.19) are both low molecular weight.

The two papers cited by the author did not explain this issue.

[12] Wang, D.; Tang, J.; Wang, Y.; Ramishetti, S.; Fu, Q.; Racette, K.; Liu, F., Multifunctional nanoparticles based on a single-molecule modification for the treatment of drug-resistant cancer. Molecular Pharmaceutics 2013, 10 (4), 1465-1469.

[13] Jeong, S. H.; Chavani, O.; Burns, K.; Porter, D.; Findlay, M.; Helsby, N., Severe 5-Fluorouracil-Associated Gastrointestinal Toxicity Unexplained by Dihydropyrimidine Dehydrogenase Deficiency and Renal Impairment: Should We Be Investigating Other Elimination Pathways to Assess the Risk of 5-Fluorouracil Toxicity? European Journal of Drug Metabolism and Pharmacokinetics 2021, 46 (6), 817-820.

2. in figure 5 legend, you need to add which range the p-value represents. Like *,p<0.05; **,p<0.01; ***,p<0.001; n.s., no significant. And the values of micelle, liposome, and LMHPTX seem close, why there are significant differences between them.

3. in figure 6, the author need to add other 2 control group, 5FU-loaded Micelles and 5FU-loaded Micelles+PTX-loaded liposomes.

4. Cytotoxicity assays on micelles, liposomes and LMH should be done.

5. in figure 6, the author need to add other 3 control group, 5FU-loaded Micelles, PTX-loaded liposomes and 5FU-loaded Micelles+PTX-loaded liposomes.

6. For PTX-loaded liposome, PTX release amount shows no difference at 48 h between 37 and 48 °C. But why in viability assay, the toxicity of PTX-loaded liposomes is increased. I would like to ask whether it is the increase in temperature itself that can kill the cells. Is it possible that the increase in the toxicity of other groups is only due to the cell death caused by the increase in temperature. And in figure7, increase of temperature, it indeed induce higher dead cells.

7. in figure7, why the PTX+5-FU shows a higher toxicity than LMH(PTX-5-FU) in hela cells. Also, there are no increase toxicity from 37 to 42 °C in LMH(PTX-5-FU)-treated Hela cells. Why the toxicity of PTX+5-FU increased when temperature increased in Neuro cells? Why the toxicity of LMH(PTX-5-FU) decreased when temperature increased in Neuro cells? Can you quantify the fluorescence in figure7?

Round 2

Reviewer 1 Report

I would like to acknowledge that the authors have made significant improvements in response to the reviewer's comments. I commend the authors for their diligence and commitment to enhancing the manuscript.

Reviewer 2 Report

Thank you for your response. I agree to publish it.